# Phase I Study to Assess the Safety and Immunogenicity of an Intradermal COVID-19 DNA Vaccine Administered Using a Pyro-Drive Jet Injector in Healthy Adults

**DOI:** 10.3390/vaccines10091427

**Published:** 2022-08-30

**Authors:** Hironori Nakagami, Hiroki Hayashi, Jiao Sun, Yuka Yanagida, Takako Otera, Futoshi Nakagami, Shigeto Hamaguchi, Hisao Yoshida, Hideo Okuno, Shota Yoshida, Ryo Nakamaru, Serina Yokoyama, Taku Fujimoto, Kazuhiro Hongyo, Yukihiro Akeda, Ryuichi Morishita, Kazunori Tomono, Hiromi Rakugi

**Affiliations:** 1Department of Health Development and Medicine, Graduate School of Medicine, Osaka University, 2-2 Yamada-oka, Suita 565-0871, Osaka, Japan; 2Division of Infection Control and Prevention, Osaka University Hospital, 2-2 Yamada-oka, Suita 565-0871, Osaka, Japan; 3Department of Geriatric and General Medicine, Graduate School of Medicine, Osaka University, 2-2 Yamada-oka, Suita 565-0871, Osaka, Japan; 4Department of Clinical Gene Therapy, Graduate School of Medicine, Osaka University, 2-2 Yamada-oka, Suita 565-0871, Osaka, Japan

**Keywords:** COVID-19, SARS-CoV-2, DNA vaccine

## Abstract

We conducted a nonrandomized, open-label phase I study to assess the safety and immunogenicity of an intradermal coronavirus disease 2019 (COVID-19) DNA vaccine (AG0302-COVID-19) administered using a pyro-drive jet injector at Osaka University Hospital between Yanagida November 2020 and December 2021. Twenty healthy volunteers, male or female, were enrolled in the low-dose (0.2 mg) or high-dose (0.4 mg) groups and administered AG0302-COVID19 twice at a 2-week interval. There were no adverse events that led to discontinuation of the study drug vaccination schedule. A serious adverse event (disc protrusion) was reported in one patient in the high-dose group, but the individual recovered, and the adverse event was not causally related to the study drug. In the analysis of the humoral immune response, the geometric mean titer (GMT) of serum anti-SARS-CoV-2 spike glycoprotein-specific antibody was low in both the low-dose and high-dose groups (246.2 (95% CI 176.2 to 344.1, 348.2 (95% CI 181.3 to 668.9)) at the 8 weeks after first vaccination. Regarding the analysis of the cellular immune, the number of IFN-γ-producing cells responsive to the SARS-CoV-2 spike glycoprotein increased with individual differences after the first dose and was sustained for several months. Overall, no notable safety issues were observed with the intradermal inoculations of AG0302-COVID19. Regarding immunogenicity, a cellular immune response was observed in some subjects after AG0302-COVID19 intradermal inoculation, but no significant antibody production was observed.

## 1. Introduction

To prevent the spread of coronavirus infection 2019 (COVID-19), an effective and safe vaccine must be widely disseminated in a short time. Many pharmaceutical companies are developing vaccines against severe acute respiration syndrome coronavirus 2 (SARS-CoV-2), including vaccines based on inactivated target virus or viral proteins, adenovirus vectors, and RNA/DNA from SARS-CoV-2 [1,2,3,4,5,6]. The major target antigen is the spike glycoprotein of SARS-CoV-2, which is essential for virus entry into cells [7]. Surprisingly, within one year, Pfizer/BioNTech and Moderna received temporary authorization for emergency use of their COVID-19 mRNA vaccines. AstraZeneca and Johnson & Johnson also later received temporary authorization for the emergency use of adenovirus vector vaccines.

DNA vaccines are one of the candidate modalities for COVID-19 vaccines, which consist of genes or fragments encoding immunogenic antigens delivered to host cells by using DNA plasmids as vectors. Antigen-presenting cells (APCs) are the major target cells that take up genetic materials, and the translated proteins or protein fragments in APCs are further processed into peptides that bind to major histocompatibility complex (MHC) class I or II for T-cell activation. Their surface exposure can result in cross-priming and presentation of antigens to both CD4 and CD8 T cells, leading to efficient humoral and cellular immune responses [8,9]. A DNA vaccine for COVID-19, ZyCoV-D, was developed for COVID-19 using a needle-free delivery device for intradermal injection, and the efficacy was 66.6% (95% CI 47.6–80.7%) with no occurrence of solicited adverse events in a phase III clinical trial, leading to temporary authorization in India [10]. 

We have also developed DNA vaccines for SARS-CoV-2 [11]. In this clinical trial, we explored the possibility of reducing the amount of plasmid DNA delivered by intradermal administration using a pyro-drive jet injector. 

## 2. Materials and Methods

### 2.1. Study Design

This study is an open-label, noncontrolled phase I study to assess the safety and immunogenicity of two doses of intradermal AG0302-COVID19 (0.2 mg or 0.4 mg) delivered using a pyro-drive jet injector in healthy adults, which was approved by the institutional review board of Osaka University Hospital (209008-B). This study was registered at the Japan Registry of Clinical Trials (jRCT) under jRCT2051200085. 

### 2.2. DNA Vaccine

AG0302-COVID19 is comprised of the DNA plasmid Vector pVAX1 carrying gene expressing spike-S protein of SARS-CoV-2. The virus RNA of SARS-CoV-2 (isolate Wuhan-Hu-1; MN_908,947.3) was obtained from the National Institute of Infectious Disease (Tokyo, Japan). A highly optimized DNA sequence encoding the SARS-CoV-2 Spike glycoprotein was created using an in silico gene optimization algorithm to enhance protein expression [11]. The plasmid DNA was propagated for larger-scale production under current Good Manufacturing Practice conditions, and finally prepared in 2 mg/mL suspended in phosphate buffer saline. 

### 2.3. Endpoints

#### 2.3.1. Primary Endpoint

Safety and tolerability: incidence of treatment-emergent adverse events (AEs), and frequency and severity of each adverse event, including solicited local and systemic AEs after the first vaccination.

Immunogenicity (Weeks 2, 4, 6, 8): change in geometric mean titer (GMT) of serum anti-SARS-CoV-2 spike (S) glycoprotein-specific antibody.

#### 2.3.2. Secondary Endpoint

Change in GMT of anti-S-specific antibody, GMT of the neutralizing activity against SARS-CoV-2 pseudovirus, and IFN-γ production in response to S by peripheral blood mononuclear cells. 

### 2.4. Eligibility Criteria

#### 2.4.1. Inclusion

(1)Subjects who voluntarily provided written consent to participate in this clinical trial.(2)Subjects whose age at the time of consent was 20 years to 65 years.(3)Subjects who were negative for SARS-CoV-2 infection by PCR test.(4)Subjects who were negative for both SARS-CoV-2 IgM antibody and SARS-CoV-2 IgG antibody by antibody test.

#### 2.4.2. Exclusion

(1)Subjects with symptoms of suspected COVID-19 (respiratory symptoms, headache, malaise, olfactory disorders, taste disorders, etc.).(2)Subjects with a history of COVID-19 (self-reported).(3)Subjects who participated in unapproved vaccine clinical trials.(4)Subjects with an axillary temperature of 37.0 degrees or higher.(5)Subjects who had a history of anaphylaxis.(6)Subjects who had a current or history of serious renal, cardiovascular, respiratory, liver, kidney, gastrointestinal, or neuropsychiatric diseases.(7)Subjects with a history of convulsions or epilepsy.(8)Subjects with a history of an immunodeficiency diagnosis.(9)Subjects who had a close relative (within third degree) with congenital immunodeficiency.(10)Subjects who had current bronchial asthma.(11)Subjects who had a fever of 39.0 °C or higher within 2 days after the last vaccination, and those who suspected allergies such as a systemic rash.(12)Females who wished to become pregnant from the date of study registration to 12 weeks after the first inoculation of the investigational drug, and pregnant females who were breast-feeding. In addition, females who may have become pregnant and their male sexual partners were encouraged to use appropriate contraceptives (pill), condoms, vasectomy, tubal ligation, diaphragms, intrauterine devices, spermicides, intrauterine hormone-releasing systems, etc., from the study entry date until 12 weeks after the first vaccination.(13)Subjects who participated in clinical trials of other unapproved drugs and received the investigational drug within 4 weeks before the start of this clinical trial (starting from the day of vaccination).(14)Subjects who received a live vaccine, inactivated vaccine, or toxoid within 4 weeks before the start of this clinical trial (starting from the day of vaccination).(15)Subjects who were administered drugs that affect the immune system (excluding external preparations) such as immunomodulators (disease-modifying antirheumatic drug, DMARDs, etc.), immunosuppressants, biologics, etc., within 4 weeks before vaccination.(16)Subjects who received blood transfusion or gamma globulin therapy within 12 weeks before vaccination, or high-dose gamma globulin therapy (200 mg/kg or higher) within 24 weeks before vaccination.(17)Subjects who had a history of overseas travel within 4 weeks before the start of the clinical trial (starting from the day of vaccination).(18)Subjects who were unable to comply with the clinical trial protocol and follow-up (for mental, family, social, or geographical reasons).(19)Subjects who were judged to be ineligible for this clinical trial by the investigator.

### 2.5. Treatment

Two dose groups (a low-dose group (0.2 mg of plasmid DNA) and a high-dose group (0.4 mg of plasmid DNA)) were established in this study. The study period for each subject consisted of a screening period, an inoculation period, an observation period, and a follow-up period. Intradermal inoculation was performed using a pyro-drive jet injector that was developed for intradermal injection of the DNA vaccine.

After consent was obtained, subjects who were screened and found eligible were enrolled. A total of 0.1 mL of the study drug was filled into an investigational device and inoculated intradermally at one site in the low-dose group. For the high-dose group, two investigational devices were filled with 0.1 mL of the study drug, and one device was used per site for intradermal inoculation at two sites. The inoculation solution was inoculated twice, 2 weeks apart, in the middle of the right or left brachial deltoid muscle. Observations, investigations, or tests to assess safety and immunogenicity were conducted 2 weeks (just before the second inoculation), 4 weeks, 6 weeks, 8 weeks, 12 weeks, 20 weeks, 24 weeks, and 52 weeks after the first dose of the study drug.

### 2.6. Statistical Analysis

For the antibody titer and neutralizing antibody, the percentage increase in GMT at 2 weeks (just before the second dose), 4 weeks, 6 weeks, 8 weeks, 12 weeks, 20 weeks, and 24 weeks after the first dose of the study drug and its two-sided 95% confidence interval were calculated overall and for each dose group, using the GMT before the first dose of the study drug as the reference. The GMT and its two-sided 95% confidence intervals at each time point were also calculated. In terms of the IFN-γ production, summary statistics were calculated for the measurements at each time point, similarly. For the safety evaluation, the number of subjects and two-sided 95% confidence intervals of adverse events occurring from the time of the first dose of study drug to 8 weeks after the first dose were calculated overall and for each dose group. Statistical analysis was performed using SAS version 9.4 (SAS Institute Inc, Cary, NC, USA). 

## 3. Results

Twenty healthy volunteers were enrolled in the low-dose (0.2 mg) or high-dose (0.4 mg) groups and administered AG0302-COVID19 twice at a 2-week interval (Table 1). From the viewpoint of subject protection, in the sentinel first case of both the low-dose group and the high-dose group, we confirmed that no serious AEs with undeniable causal relationships to the study drug inoculation were observed within 24 h after the injection. After that, the study proceeded to the second and subsequent cases. In addition, the first case of the high-dose group was started after the safety was confirmed for 7 days after the completion of the first inoculation of all subjects in the low-dose group.

There were no AEs that led to discontinuation of the study drug vaccination (Table 2). From 8 weeks after the first dose of the study drug to 52 weeks after the first dose (follow-up period), one serious AE (disc protrusion) was reported in one patient in the high-dose group, but the individual recovered, and the AE was not causally related to the study drug. AEs occurred in 17 of 20 patients (85.0%) (7 patients (70.0%) and 10 patients (100%)) in the low-dose and high-dose groups, respectively. Of the AEs that occurred from the time of the first dose of the study drug to the 28th day after the last dose, AEs other than specific AEs (local and systemic reactions) that occurred up to 2 weeks after each dose of the study drug occurred in 11 individuals (55.0%) (four (40.0%) and seven (70.0%), respectively), which included nasopharyngitis and bradycardia, abdominal pain, nausea, fatigue, injection site rash, vaccination site pruritus, internal bleeding at the vaccination site, abnormal liver function, gingivitis, hypoglycemia, arthralgia, back pain, disc protrusion, headache, somnolence, cough, and pruritus in one case each (5.0%). The causal relationship of disc protrusion to the study drug was re-evaluated when it was reported as a serious AE during the follow-up period, and it was found to be not related. In addition, no consistent trend was observed in laboratory values and vital signs. In terms of intradermal inoculation using a pyro-drive jet injector, no investigational device failures were reported.

Regarding the SARS-CoV-2 spike (S) glycoprotein-specific antibody titer before the first dose, all subjects in the low-dose group had antibody titers below the cutoff point, and three subjects in the high-dose group had antibody titers above the cutoff point. Antibody titers at 2, 4, 6, and 8 weeks after the first dose were above the cutoff point in two subjects in the low-dose group (two time points in total) and in three subjects in the high-dose group (12 time points in total). Two subjects in the low-dose group (three time points in total) and four subjects in the high-dose group (five time points in total) had antibody titers above the cutoff point at 12, 20, 24, and 52 weeks after the first dose of the study drug (Table 3). The percent change in GMT at 12, 20, and 24 weeks after the first dose of the study drug was 1.1, 1.0, 1.1, and 2.8 in the low-dose group and 0.9, 0.6, 0.7, and 1.4 in the high-dose group, respectively. The GMT of neutralizing activity (ID50) against SARS-CoV-2 pseudoviruses at each time point after inoculation was also low in both the low-dose and high-dose groups (Table 4).

Regarding the analysis of cellular immunity, the number of IFN-γ-producing cells responsive to SARS-CoV-2 spike (S) glycoprotein was increased after inoculation (Table 5). The mean (standard deviation) of the change in the number of IFN-γ-producing cells at each time point after inoculation was 16.75 (43.09), 23.50 (38.75), 19.75 (49.76), 25.00 (59.81), 5.75 (26.54), 12.75 (22.28), and 19.00 (40.88) in the low-dose group. Similarly, the mean number of IFN-γ-producing cells at each time point was 7.50 (15.63), 55.75 (121.34), 21.25 (44.13), 26.00 (54.09), 1.50 (17.37), 23.06 (33.42), and 18.61 (31.53) in the high-dose group.

## 4. Discussion

Regarding the history of DNA vaccines, the first clinical trial started in the 1990s [12,13], and many clinical trials that focus on DNA vaccination have been registered. Overall, no severe AEs related to the DNA vaccine were reported in the safety evaluation [12,13,14]. A few DNA-based vaccines have been approved for veterinary use, including vaccines against West Nile virus in horses [15] and canine melanoma [16]. In terms of plasmid DNA vaccines for COVID-19 in humans, INOVIO developed a DNA vaccine that was intradermally delivered by electroporation [17], and Zydus also developed a plasmid DNA vaccine, ZyCoV-2, which are both intradermally delivered by bioinjectors under temporary authorization in India [9,18].

This first-in-human phase 1/2 study of intradermal AG0302 delivered using a pyro-drive jet injector was carried out in an intensive observational unit with monitoring for 56 weeks. Each vaccination was followed by frequent safety follow-up with subjects until 28 days after the last dose of vaccine. In this study, the intradermal DNA vaccine was well-tolerated in the twenty healthy adults in two groups, with no vaccine-related severe AEs. These findings correlate with the previous clinical evaluations of DNA vaccine candidates that were reported to be safe and well-tolerated in healthy subjects [9,12,13,14,17,18].

The humoral immune response to intradermal AG0302 was low, especially in the low-dose group. Although the exact mechanism of differences in responses among species is still unknown, the dose of our vaccine (0.2 mg or 0.4 mg) was lower than that of the ZyCoV-2 vaccine (1 mg or 2 mg), as humoral responses were relatively low in the low-dose (1 mg) group of the ZyCoV-2 vaccine compared with those in the high-dose (2 mg) [18]. They intradermally administered a maximum of 2 mg of DNA vaccine three times (total 6 mg), which was much higher than our maximum (total 0.8 mg). Thus, the low humoral response might be due to insufficient antigen expression in our clinical trials due to a difference in dosage and administration. Regarding the cellular immune response, which was assessed by spike protein-stimulated IFN-γ expression in an ELISPOT assay, immune reactions were observed in both the low- and high-dose groups. Although the sample size was too small and the results included large individual differences, the scores of IFN-γ production were comparable with the results in the clinical study of other COVID-19 vaccines [19]. As the potential of DNA vaccines for induction of cellular immunity has already been reported in previous clinical trials, the results of our clinical trial display the same trend [9,12,13]. Based on lessons from the success of the mRNA vaccines, low immunogenicity of mRNA achieved by modification with psudouridine and combination with lipid nanoparticles for immuno-modulation might be key to activate the human immune system [20,21]. This approach may reveal directions to improve the DNA vaccine to overcome differences in the immune system among species. We will continuously optimize the DNA vaccine.

This study includes the following study limitation. In this study, until week 24, only one participant in the high-dose group was discontinued. After that, an mRNA vaccine for COVID-19 was approved and started to be administered in Japan. In this protocol, the participants must stop this study to receive an mRNA vaccine, which was also recommended to control this viral infection as a society. Therefore, until week 56, 13 participants (8 in the low-dose group and 5 in the high-dose group) decided to stop this study to receive an mRNA vaccine. Therefore, most of the participants did not complete this study up to week 56. In terms of efficiency evaluation, the results of immune response peaked before week 24, and the results can be sufficient to draw the conclusion for efficiency evaluation. A safety evaluation was conducted up to and including week 56, not only for the remaining six participants but also other participants by phone or email. All AEs recovered by week 24, and the results can be also effective for the conclusion. For the further evaluation after week 24, six participants (two in the low-dose group and four in the high-dose group) were evaluated in week 56. However, the possibility of an increase in antibody titer due to subclinical infection could not be ruled because of the overlap with the period of infection spread in Japan. Thus, the results of immune response for week 56 were excluded from the efficacy evaluation.

The delivery and transfection system has been optimized for DNA vaccines in human clinical trials as well as animal studies. Actually, jet injectors [22] and electroporation [23] have enhanced responses through an increased efficiency of DNA delivery and have been recently utilized in clinical trials. The use of a needleless injection system for vaccine administration can result in a reduction in the occurrence of side effects typically associated with needle injection, such as injection-site pain. As shown in Table 2, there were only a few cases of injection-site pain and swelling with no device failures. In this study, the safety of a pyro-drive jet injector was evaluated, but the optimization for an efficient immune response will be further explored not only for DNA vaccines but also for other vaccine modalities. The plasmid DNA can be stored at 2–8 °C and at 25 °C for a few months. The thermostability of the DNA vaccine will aid transportation and storage of the vaccine and reduce any cold chain breakdown resulting in vaccine waste, which might be an advantage of DNA vaccines among several modalities of vaccines. 

## 5. Conclusions

No notable safety issues were observed with the two intradermal inoculations of AG0302-COVID19. Regarding immunogenicity, an immune response was observed in some subjects after AG0302-COVID19 intradermal inoculation, but no significant antibody production was observed. Although our approach was not successful in this clinical trial, we will continue to iterate based on this experience. In the future, we hope that DNA vaccines will be successfully modified to overcome the problem of differences in the immune system among species. 

## Figures and Tables

**Table 1 vaccines-10-01427-t001:** Basic characteristics.

	Low Dose (0.2 mg)	High Dose (0.4 mg)
	N = 10	N = 10
Female/male	4/6	5/5
Age	48.6 ± 10.6	44.2 ± 15.3
Height	163.3 ± 8.6	166.5 ± 8.9
Weight	56.2 ± 9.7	58.6 ± 9.4
Medical history	1	1
Underlying diseases	9	10
Smoking history/current	1/1	3/2

**Table 2 vaccines-10-01427-t002:** Treatment-emergent adverse events.

	Low (0.2 mg)	High (0.4 mg)	Total
	N = 10	N = 10	N = 20
	n [95% CI]	n [95% CI]	n [95% CI]
Treatment-emergent adverse event	7 [34.8, 93.3]	10 [69.2, 100.9]	17 [62.1, 96.8]
Blood cell disorders (anemia)	1 [0.3, 44.5]	0 [0.0, 30.8]	1 [0.1, 24.9]
Cardiac disorders (bradycardia)	0 [0.0, 30.8]	1 [0.3, 44.5]	1 [0.1, 24.9]
Gastrointestinal disorders	3 [6.7, 65.2]	2 [2.5, 55.6]	5 [8.7, 49.1]
Abdominal pain	1 [0.3, 44.5]	0 [0.0, 30.8]	1 [0.1, 24.9]
Diarrhea	3 [6.7, 65.2]	2 [2.5, 55.6]	5 [8.7, 49.1]
Nausea	1 [0.3, 44.5]	0 [0.0, 30.8]	1 [0.1, 24.9]
General disorders and administration site conditions	4 [12.2, 73.8]	8 [44.4, 97.5]	12 [36.1, 80.9]
Fatigue	0 [0.0, 30.8]	1 [0.3, 44.5]	1 [0.1, 24.9]
Malaise	0 [0.0, 30.8]	2 [2.5, 55.6]	2 [1.2, 31.7]
Vaccination site erythema	1 [0.3, 44.5]	3 [6.7, 65.2]	4 [5.7, 43.7]
Vaccination site rash	1 [0.3, 44.5]	0 [0.0, 30.8]	1 [0.1, 24.9]
Vaccination site induration	1 [0.3, 44.5]	1 [0.3, 44.5]	2 [1.2, 31.7]
Vaccination site pain	2 [2.5, 55.6]	2 [2.5, 55.6]	4 [5.7, 43.7]
Vaccination site itching	0 [0.0, 30.8]	1 [0.3, 44.5]	1 [0.1, 24.9]
Vaccination site internal hemorrhage	0 [0.0, 30.8]	1 [0.3, 44.5]	1 [0.1, 24.9]
Vaccination site swelling	1 [0.3, 44.5]	0 [0.0, 30.8]	1 [0.1, 24.9]
Hepatobiliary disorders (abnormal hepatic function)	0 [0.0, 30.8]	1 [0.3, 44.5]	1 [0.1, 24.9]
Infections and infestations	2 [2.5, 55.6]	4 [12.2, 73.8]	6 [11.9, 54.3]
Cystitis	0 [0.0, 30.8]	2 [2.5, 55.6]	2 [1.2, 31.7]
Periodontitis	1 [0.3, 44.5]	0 [0.0, 30.8]	1 [0.1, 24.9]
Nasopharyngitis	1 [0.3, 44.5]	1 [0.3, 44.5]	2 [1.2, 31.7]
Pyuria	0 [0.0, 30.8]	1 [0.3, 44.5]	1 [0.1, 24.9]
Investigations	1 [0.3, 44.5]	2 [2.5, 55.6]	3 [3.2, 37.9]
High blood pressure	0 [0.0, 30.8]	2 [2.5, 55.6]	2 [1.2, 31.7]
Blood uric acid increased	1 [0.3, 44.5]	0 [0.0, 30.8]	1 [0.1, 24.9]
Metabolism and nutrition disorders (hypoglycemia)	0 [0.0, 30.8]	1 [0.3, 44.5]	1 [0.1, 24.9]
Musculoskeletal and connective disorders	1 [0.3, 44.5]	2 [2.5, 55.6]	3 [3.2, 37.9]
Joint pain	1 [0.3, 44.5]	0 [0.0, 30.8]	1 [0.1, 24.9]
Back pain	0 [0.0, 30.8]	1 [0.3, 44.5]	1 [0.1, 24.9]
Disc protrusion	0 [0.0, 30.8]	1 [0.3, 44.5]	1 [0.1, 24.9]
Nervous system disorders	2 [2.5, 55.6]	4 [12.2, 73.8]	6 [11.9, 54.3]
Headache	2 [2.5, 55.6]	4 [12.2, 73.8]	6 [11.9, 54.3]
Somnolence	1 [0.3, 44.5]	0 [0.0, 30.8]	1 [0.1, 24.9]
Respiratory, thoracic, and mediastinal disorders (cough)	1 [0.3, 44.5]	0 [0.0, 30.8]	1 [0.1, 24.9]
Skin and subcutaneous tissue disorders (itching)	0 [0.0, 30.8]	1 [0.3, 44.5]	1 [0.1, 24.9]

**Table 3 vaccines-10-01427-t003:** GMT of SARS-CoV-2 S glycoprotein-specific antibody.

	Low Dose (0.2 mg)	High Dose (0.4 mg)
	N = 10	N = 10
	GMT [95% CI]	GMT [95% CI]
Week 0	200 [-, -]	324.9 [182.8, 577.4]
Week 2 (day 28)	200 [-, -]	303.1 [187.8, 489.4]
Week 4 (day 56)	200 [-, -]	324.9 [182.8, 577.4]
Week 6 (day 84)	200 [-, -]	324.9 [182.8, 577.4]
Week 8 (day 112)	246.2 [176.2, 344.1]	348.2 [181.3, 668.9]
Week 12 (day 140)	229.7 [167.9, 314.4]	303.1 [177.9, 516.6]
Week 20 (day 196)	200 [-, -]	200 [-, -] (N = 9)
Week 24 (day 224)	214.4 [183.2, 250.7]	252.0 [147.9, 429.3] (N = 9)

**Table 4 vaccines-10-01427-t004:** GMT of neutralizing activity (ID50) against SARS-CoV-2 pseudoviruses.

	Low Dose (0.2 mg)	High Dose (0.4 mg)
	N = 10	N = 10
	GMT [95% CI]	GMT [95% CI]
Week 0	1.0 [-, -]	1.6 [0.9, 2.9]
Week 2 (day 28)	1.0 [-, -]	1.5 [0.6, 3.7]
Week 4 (day 56)	1.3 [0.8, 2.0]	7.6 [1.5, 37.3]
Week 6 (day 84)	1.5 [0.9, 2.6]	2.6 [1.1, 6.4]
Week 8 (day 112)	1.0 [-, -]	5.5 [1.4, 21.3]
Week 12 (day 140)	1.0 [-, -]	2.1 [1.0, 4.5]
Week 20 (day 196)	1.0 [-, -]	2.1 [0.8, 6.1] (N = 9)
Week 24 (day 224)	1.0 [-, -]	2.2 [0.9, 5.2] (N = 9)

**Table 5 vaccines-10-01427-t005:** Cellular response (IFN-γ) to AG0302.

	Low Dose (0.2 mg)	High Dose (0.4 mg)
	(N = 10)	(N = 10)
Week 0	5.50 ± 9.34	6.00 ± 12.03
Week 2 (day 28)	22.25 ± 44.48	13.50 ± 17.80
Week 4 (day 56)	29.00 ± 41.22	61.75 ± 120.38
Week 6 (day 84)	25.25 ± 51.82	27.25 ± 43.48
Week 8 (day 112)	30.50 ± 62.80	32.00 ± 50.48
Week 12 (day 140)	11.25 ± 28.73	7.50 ± 10.21
Week 20 (day 196)	18.25 ± 19.93	25.56 ± 32.37 (N = 9)
Week 24 (day 224)	24.50 ± 43.25	21.11 ± 30.98 (N = 9)

## Data Availability

Data will be made available on reasonable request to the corresponding author.

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
