# Peer review of "Phase I Study to Assess the Safety and Immunogenicity of an Intradermal COVID-19 DNA Vaccine Administered Using a Pyro-Drive Jet Injector in Healthy Adults"

_vaccines, 2022, doi:10.3390/vaccines10091427_

Round 1

Reviewer 1 Report

Nakagami et al., conducted a nonrandomized, open-label phase I/II study to assess the safety and immunogenicity of AG0302-COVID-19 DNA vaccine administered via intradermal injection using pyro-drive jet injector. The subjects enrolled (20) were immunized with low (0.2 mg) and high dose (0.4 mg) at 2 weeks interval. They observe no serious side effect except for one subject who eventually recuperated. Unfortunately, there was low induction of anti-SARS-CoV-2 spike glycoprotein specific antibody and neutralizing antibody in the serum, however, a sustainable number of SARSCoV-2 spike glycoprotein specific IFN-γ-producing cells increased within the population after the first dose.

Major comments

1.     The number of subjects is considered small. The number might be suitable for Phase I but not suitable for phase II. The authors should justify their number of subjects. Given that most of the participants did not complete the study, it might be very difficult to arrive at a conclusion on the safety and efficacy of the vaccine.  

2.     Were the causes of the AEs investigated? Why was the study continued if 17 out of 20 had AEs? The statistical interpretation of this data (Table 2) should be provided.

3.     There was no negative control (Placebo) in this study. Also, for the antibody induced investigation, the author should have also use human convalescent COVID-19 sera and compare especially with the IFN-γ. This will further support that the vaccine could increase the amount of IFN-γ production.

Minor comments

1.     It is not clear with the abstracts the number of dosages used and the doses given at each time of immunization. Is the 0.2 mg and 0.4 mg doses for different groups or one group received 0.2 mg as prime and 0.4 mg as boost? Please clarify.

2.     2.2 (First line) “AG0302-COVID19 is comprised of d DNA plasmid Vector pVAX1” change ‘d’ to ‘the’

3.     How is this vaccine platform different and/or better than the already in use DNA vaccine for COVID-19 e.g. the Zydus or INOVIO as mentioned by the authors?

Author Response

Thank you very much for your very important comments. According to your suggestions, we have certainly tried our best to revise the manuscript.

1. The number of subjects is considered small. The number might be suitable for Phase I but not suitable for phase II. The authors should justify their number of subjects. Given that most of the participants did not complete the study, it might be very difficult to arrive at a conclusion on the safety and efficacy of the vaccine.

We appreciate your comments. The primary endpoint of this study is safety evaluation because it is first-in-human study for intradermal COVID-19 DNA vaccine administered using a pyro-drive jet injector. We conducted twenty subjects in low-dose and high-dose groups. As you suggested, the study size is small for phase II. Thus, we changed the title and description of manuscript into “ Phase I study to assess the safety and immunogenicity of an intradermal COVID-19 DNA vaccine administered using a pyro-drive jet injector in healthy adults”, and deleted the immunogenicity evaluation from the primary endpoint.  

We have following explanation about that most of the participants did not complete study. Until week 24, only one participant in the high-dose group was discontinued. After that, mRNA vaccine for COVID-19 was approved and started to be administered in Japan, which was also recommended to control this viral infection as a society. In this protocol, the participants must stop this study to receive mRNA vaccine. Until week 56, 13 participants (8 in the low-dose group and 5 in the high-dose group) decided stopping this study to receive mRNA vaccine. Therefore, most of the participants did not complete this study until week 56. In terms of efficiency evaluation, the results of immune response have been peaked out before week 24, and the results can be sufficient to draw the conclusion for efficiency evaluation. Safety evaluation was conducted up to and including week 56 not only for the remaining 6 participants but also other participants by phone or email. All AEs have been recovered until week 24, the results can be also effective for the conclusion.

We described this explanation in Discussion.

  1. Were the causes of the AEs investigated? Why was the study continued if 17 out of 20 had AEs? The statistical interpretation of this data (Table 2) should be provided.

We appreciate your comments. There were no AEs that led to discontinuation of the study drug vaccination. From 8 weeks to 52 weeks after the first dose of the study drug (follow-up period), one serious AE (disc protrusion) was reported in one patient in the high-dose group, but the individual recovered and the AE was not causally related to the study drug. AEs included local reaction (i.e., vaccination site erythema, induration and pain) which are also recovered soon. In addition, according to your suggestion, the statistical analysis (95% CI) was performed in table 2.

  1. There was no negative control (Placebo) in this study. Also, for the antibody induced investigation, the author should have also use human convalescent COVID-19 sera and compare especially with the IFN-γ. This will further support that the vaccine could increase the amount of IFN-γ production.

We appreciate your comments. We obtained human convalescent COVID-19 serum as a candidate International Standard for anti-SARC-CoV-2 antibody (NIBSC code 20/136). It is a freeze-dried preparation of a pool of plasma from 11 SARS-CoV-2 recovered patients from England, UK. Material was collected more than 28 days after the onset of symptoms. The antibody titer of this human convalescent COVID-19 sera was 3,200, that is more than 10 times higher than that of our results in this study. However, IFN-γ production is measured by utilizing PBMC (peripheral blood mononuclear cells) which cannot be obtained as international standard samples. In the literature of clinical study to evaluate the combination vaccine with adenovirus vaccine and mRNA vaccine (Lancet. 2021 Sep 4;398(10303):856-869), cellular response was evaluated with IFN-γ production. In this study, the scores of IFN-γ production were 48 for adenovirus vaccine and 80 for mRNA vaccine, that were comparable with our results. Therefore, we concluded that a cellular response, but not antibody production, was observed in some subjects after AG0302-COVID19 intradermal inoculation. We briefly described this explanation in Discussion.

Minor comments

  1. It is not clear with the abstracts the number of dosages used and the doses given at each time of immunization. Is the 0.2 mg and 0.4 mg doses for different groups or one group received 0.2 mg as prime and 0.4 mg as boost? Please clarify.

We appreciate your comments. We first started the low dose (0.2 mg) vaccination at a 2-week interval. From the viewpoint of subject protection, the first dose of the high-dose (0.4 mg) group was started after the safety was confirmed for 7 days after the completion of the first inoculation of all subjects in the low-dose group. Accordingly, we changed the description into “Twenty healthy volunteers, male or female, were enrolled in the low-dose (0.2 mg) or high-dose (0.4 mg) groups and administered AG0302-COVID19 twice at a 2-week interval”.

  1. 2 (First line) “AG0302-COVID19 is comprised of d DNA plasmid Vector pVAX1” change ‘d’ to ‘the’

We apologized the type and changed it into “the DNA plasmid Vector pVAX1”

  1. How is this vaccine platform different and/or better than the already in use DNA vaccine for COVID-19 e.g. the Zydus or INOVIO as mentioned by the authors?

We appreciate your comments. If we compare our DNA vaccine to theirs (Zydus or INOVIO), plasmid DNA backbone (pVAX1) and antigen (Spike protein) were same, however the concentration of the formulation was five times higher than ours. They intradermally administered 2mg of DNA vaccine three times (total 6mg) which was much higher than our maximum (total 0.8 mg). In addition, Zydus utilized the jet injector for intradermal injection and INOVIO utilized the electroporator to enhance the immune response that were optimized for human clinical trials. There was a difference in dosage and administration, and the optimization of pyro-drive jet injector for an efficient immune response will be further explored for human clinical trial. We briefly described this explanation in Discussion.

Reviewer 2 Report

   Hironori and colleagues have performed an interesting study for DNA vaccine on COVID-19. However, there are still something issues need to be addressed before published.

1.     There are no any statistical data in Abstract. For example, authors should report the geometric mean titer with 95% CI in Abstract.

2.     As stated in the Methods, the authors actively collect the information on AEs. Please clarify if there is any active monitoring tool or mechanism ensuring the data collection, including classification of possible AEs and reporting.

3.     The limitations in method could be move to Discussion.

4.     Please add statement to demonstrate how to decide the sample size in the present study.

5.     The statistical analysis method should be added in the Methods.

Author Response

Thank you very much for your very important comments. According to your suggestions, we have certainly tried our best to revise the manuscript.

  1. There are no any statistical data in Abstract. For example, authors should report the geometric mean titer with 95% CI in Abstract.

We appreciate your comments. We added the GMT with 95% CI in Abstract as follows, “the geometric mean titer (GMT) of serum anti-SARS-CoV-2 spike glycoprotein specific antibody was low in both the low-dose and high-dose groups [246.2 (95% CI 176.2 to 344.1, 348.2 (95% CI 181.3 to 668.9)] at 8 weeks after first vaccination.”

  1. As stated in the Methods, the authors actively collect the information on AEs. Please clarify if there is any active monitoring tool or mechanism ensuring the data collection, including classification of possible AEs and reporting.

We appreciate your comments. In this study, we followed up the participants for almost one year after the first vaccination. We have nine visits for efficiency and safety evaluation, and the research coordinators also collected the information by phone or email if the participants have any accidents or AEs until recovery.

  1. The limitations in method could be move to Discussion.

We appreciate your comments. According to your suggestion, we described the study limitation in Discussion section.

  1. Please add statement to demonstrate how to decide the sample size in the present study.

We appreciate your comments. The primary endpoint of this study is safety evaluation because it is first-in-human study for intradermal COVID-19 DNA vaccine administered using a pyro-drive jet injector. Therefore, we conducted twenty subjects in low-dose and high-dose groups. Based on these initial results of immune response, we will be able to determine the suitable dose and calculate the sample size for the next phase. However, the study size is small for phase II. Thus, we changed the title and description of manuscript into “ Phase I study to assess the safety and immunogenicity of an intradermal COVID-19 DNA vaccine administered using a pyro-drive jet injector in healthy adults”, and deleted the immunogenicity evaluation from the primary endpoint.

  1. The statistical analysis method should be added in the Methods.

We appreciate your comments. We added the following description in Method section. “For the antibody titer and neutralizing antibody, the percentage increase in GMT at 2 weeks (just before the second dose), 4 weeks, 6 weeks, 8 weeks, 12 weeks, 20 weeks, and 24 weeks after the first dose of the study drug and its two-sided 95% confidence interval were calculated for the overall and each dose group, using the GMT before the first dose of the study drug as the reference. The GMT and its two-sided 95% confidence intervals at each time point were also calculated. In terms of the IFN-γ production, summary statistics were calculated for the measurements at each time point, similarly. For the safety evaluation, the number of subjects and two-sided 95% confidence intervals of adverse events occurring from the time of the first dose of study drug to 8 weeks after the first dose were calculated for the overall and each dose group.”

Round 2

Reviewer 1 Report

Authors essentially addressed my comments properly. no further comments.

Reviewer 2 Report

The manuscript has been well revised. Here are some minor points: 

1. Please check the number of subheadings in materials and methods.

2. Please clarify the statistical software used in statistical analysis.

Author Response

Response to Reviewer #2 (vaccines-1851561)

Thank you very much for your very important comments. According to your suggestions, we have certainly tried our best to revise the manuscript.

  1. Please check the number of subheadings in materials and methods

We appreciate your comments. Accordingly, we revised the subheadings in materials and methods

  1. Please clarify the statistical software used in statistical analysis

We appreciate your comments. Accordingly, we added the statistical software in statistical analysis.

In addition, the revised manuscript has been checked by a native English speaker.